

**Strong light scattering of highly oxygenated organic aerosols impacts significantly on**
**visibility degradation**
**Li Liu[1], Ye Kuang[2,3]\*, Miaomiao Zhai[2,3], Biao Xue[2,3], Yao He[2,3], Jun Tao[2,3], Biao Luo[2,3],**
**Wanyun Xu[4], Jiangchuan Tao[2,3], Changqin Yin[1,7], Fei Li[1,5], Hanbing Xu[6], Tao Deng[1], Xuejiao**
**Deng[1], Haobo Tan[1], Min Shao[2,3]**
[1] Institute of Tropical and Marine Meteorology, China Meteorological Administration, Guangzhou,
510640,China
[2] Institute for Environmental and Climate Research, Jinan University, Guangzhou, China.
[3] Guangdong-Hongkong-Macau Joint Laboratory of Collaborative Innovation for Environmental
Quality, Guangzhou, China.
[4] State Key Laboratory of Severe Weather & Key Laboratory for Atmospheric Chemistry, Institute of
Atmospheric Composition, Chinese Academy of Meteorological Sciences, Beijing, 100081, China
[5] Xiamen Key Laboratory of Straits Meteorology, Xiamen Meteorological Bureau, Xiamen, 361012,
China
[6] Experimental Teaching Center, Sun Yat-Sen University, Guangzhou 510275, China
[7] Shanghai Key Laboratory of Meteorology and Health, Shanghai Meteorological Bureau, Shanghai
200030, China
*Correspondence to: Ye Kuang (kuangye@jnu.edu.cn)









**Abstract**

Secondary organic aerosols (SOA) account for a large fraction of atmospheric aerosol mass and play significant roles in visibility impairment by scattering solar radiation. However, comprehensive evaluations of SOA scattering abilities under ambient relative humidity (RH) conditions on the basis of field measurements are still lacking due to the difficulty of simultaneously direct quantifications of SOA scattering efficiency in dry state and SOA water uptake abilities. In this study, field measurements of aerosol chemical and physical properties were conducted in Guangzhou winter (lasted about three months) using a humidified nephelometer system and aerosol chemical speciation monitor. A modified multilinear regression model was proposed to retrieve dry state mass scattering efficiencies (MSE, defined as scattering coefficient per unit aerosol mass) of aerosol components. The more oxidized oxygenated organic aerosol (MOOA) with O/C ratio of 1.17 was identified as the most efficient light scattering aerosol component. On average, 34% mass contribution of MOOA to total submicron organic aerosol mass contributed 51% of dry state organic aerosol scattering. Organic aerosol hygroscopicity parameter $\kappa_{OA}$ was quantified through hygroscopicity closure. The highest water uptake ability of MOOA among organic aerosol factors was revealed with $\kappa_{MOOA}$ reaching 0.23, thus further enhanced the fractional contribution of MOOA in ambient organic aerosol scattering. Especially, scattering abilities of MOOA was found to be even higher than that of ammonium nitrate under RH of <70% which was identified as the most efficient inorganic scattering aerosol component, demonstrating that MOOA had the strongest scattering abilities in ambient air (average RH of 57%) during Guangzhou winter. During the observation period, secondary aerosols contributed dominantly to visibility degradation (~70%) with substantial contributions from MOOA (16% on average), demonstrating significant impacts of MOOA on visibility degradations. Findings of this study demonstrate that more attentions need to be paid to SOA property changes in future visibility improvement investigations. Also, more comprehensive studies on MOOA physical properties and chemical formation are needed to better parameterize its radiative effects in models and implement targeted control strategies on MOOA precursors for visibility improvement.



## 1 Introduction

Atmospheric aerosols directly scatter and absorb solar radiation thus have significant radiative effects on both Earth-Atmosphere radiative budget and atmospheric environment. Aerosols represent the dominant contributor to atmospheric visibility impairment in polluted regions (Liu et al., 2017a). With the rapid industrialization and urbanization, China has been experienced severe haze pollution in recent ten years and frequent low visibility events have aroused public attention and concern, especially since 2013. In recent years, the Chinese government has implemented stringent control policies called "blue sky actions" to lower aerosol mass concentration and improve atmospheric visibility. However, Xu et al. (2020) revealed that the less than expected visibility improvement in southern China, especially the poor visibility improvement in Pearl River Delta region, due to the non-linear responses of visibility improvement to $PM_{2.5}$ (particulate matter with aerodynamic diameter less than 2.5 $\mu$m) mass concentration reduction. Several recent literature reports also proved that visibility was less improved than $PM_{2.5}$ mass concentrations. Results of Liu et al. (2020) demonstrated that increased aerosol extinction efficiency associated with nitrate was responsible for the less improved visibility in eastern China. Hu et al. (2021) raised the challenge of visibility improvement due to increased nitrate contribution in Beijing area. However, results of Xu et al. (2020) demonstrate that this situation was likely associated with both increased aerosol scattering efficiency and aerosol hygroscopicity and particularly pointed out that other than changes of inorganic aerosol components, special attention should be paid to scattering efficiency and hygroscopicity changes of secondary organic aerosol (SOA).

Organic aerosols including primary and secondary organic aerosols (POA and SOA) represent a large and sometimes even dominant fraction of submicron aerosol mass (Jimenez et al., 2009). Especially, SOA was found to contribute dominantly to total organic aerosol mass under polluted hazy conditions (Huang et al., 2014;Kuang et al., 2020a). Wang et al. (2019b) reported increased contributions of both secondary organic and inorganic aerosol mass across China due to clean air actions, and the nonlinear responses of secondary aerosol mass concentration to emission reductions were further confirmed during COVID lock down period as reported by Huang et al. (2020). Xu et al. (2019) also reported substantial changes of SOA properties such as enhanced oxidation state. However, most previous studies only paid attention to influences of nitrate increase on visibility degradation, whereas synergistic effects of SOA on visibility has never been the focus due to the complexity of





SOA hygroscopicity and scattering efficiency. Organic aerosol evolves in the atmosphere including
their sizes and chemical structures thus also their optical properties and hygroscopicity (Jimenez et al.,
2009), leading to the difficulty of quantifying contributions of organic aerosol in visibility degradation.
Li et al. (2022) reported that nitrate and SOA dominated particle extinction in dry state in Beijing due
to clean air actions, however, lacking evaluations in ambient air, stressing further the importance of
comprehensive evaluations of SOA scattering abilities under ambient relative humidity (RH)
conditions to better elucidate roles of SOA in visibility degradation and long-term visibility changes.

In this study, we comprehensively quantified the dry state mass scattering efficiencies (MSEs) of

both primary and secondary organic aerosol components and organic aerosol hygroscopicity, thus
systematically evaluated contributions of SOA factors to aerosol scattering and visibility degradation
in ambient air.
**2 Materials and Methods**
**2.1 Campaign information**

Aerosol physical and chemical properties were simultaneously measured during winter from $13^{th}$

December 2020 to $25^{th}$ February 2021 in Guangzhou urban area. Instruments were housed in an air-
conditioned container in Haizhu wetland park (Sect.S1 for site information). A $PM_{2.5}$ inlet (BGI, SCC
2.354) was used for aerosol sampling, and sample flow of 8-9 L was maintained during the observation
period thus generally satisfying the flow requirement (8 L) of 2.5 μm cutting diameter. A Nafion drier
was used to lower sample RH to less than 40%. A humidified nephelometer system with a total flow
of about 5 L/min and a quadrupole-Aerosol Chemical Speciation Monitor Q-ACSM with a flow of
3L/min were placed downstream of the drier to measure aerosol scattering abilities under controlled
RH conditions and aerosol chemical compositions. An AE33 aethalometer (Drinovec et al., 2015) set
up with a flow rate of 5 L/min was separately operated under another inlet ($PM_{2.5}$, BGI SCC 1.829) to
measure aerosol absorptions thus indirectly black carbon (BC) mass concentration. Measurements of
meteorological parameters such as temperature, wind speed and direction, and RH were made by an
automatic weather station.
**2.2 Humidified nephelometer measurements**.

The humidified nephelometer system is a laboratory-assembled one, including two Aurora 3000

nephelometers, one measuring aerosol scattering abilities (aerosol scattering and back scattering
coefficients at 450, 525 and 635 nm) under low RH conditions (mostly less than 30%, dry





nephelometer) and another one measuring aerosol scattering abilities under controlled RH conditions
(wet nephelometer). The humidified nephelometer system can operate either in fixed-RH mode or in
scanning RH mode, details about techniques of fixed RH mode and scanning RH mode were
introduced in detail in several previous studies (Kuang et al., 2017;Kuang et al., 2020b). In this study,
the humidified nephelometer system was operated in scanning RH mode before 26[th] January 2021 and
in fixed RH mode (80% RH) from 26[th] January to 9[th] February. The RH ranges of scanning RH mode
were 75-90% from 11[th] December 2020 to 5[th] January 2021 and 60-90% from 13[th] to 26[th] January 2021.
The humidified nephelometer system provides direct measurements of aerosol light scattering
enhancement factor $f(\mathrm{RH}, \lambda) = \frac{\sigma_{sp}(RH,\ \lambda)}{\sigma_{sp}(dry\ \lambda)}$ where $\sigma_{sp}(RH,\ \lambda)$ is the aerosol scattering coefficient of
light wavelength λ at condition of RH (Titos et al., 2016;Zhao et al., 2019a), and $f(\mathrm{RH}, 525)$ referred
directly to as $f(\mathrm{RH})$ was usually used to derive the optical hygroscopicity parameter $\kappa_{sca}$ through
$f(\mathrm{RH}) = 1 + \kappa_{sca} \times \frac{RH}{100-RH}$ (Brock et al., 2016;Kuang et al., 2017;Kuang et al., 2018;Xu et al.,
2020;Kuang et al., 2020b;Kuang et al., 2021b). The nephelometer measurements are associated with
truncation error and non-ideality of light source. The dry state aerosol scattering coefficients were
corrected using the empirical formula provided by Qiu et al. (2021). $RH_0$ in the dry nephelometer was
in the range of 6-49% with an average of 22%, thus dry state aerosol scattering coefficient at 525 nm
( $\sigma_{sp,525}$ ) was further corrected using measured aerosol optical hygroscopicity through
$\sigma_{sp,525} = \sigma_{sp,525,measured}/(1 + \kappa_{sca} \times \frac{RH_0}{100-RH_0})$.

**140      2.3 Q-ACSM measurements and PMF analysis.**

The Q-ACSM was deployed to routinely characterize and measure the mass concentrations of
non-refractory submicron aerosol components at a time resolution of 15min, including organics,
sulfate, nitrate, ammonium and chloride, details about Q-ACSM set-up please refer to Ng et al. (2011).
The mass concentrations and mass spectra were processed using ACSM data analysis software (ACSM
Local 1.5.10.0 Released July 6, 2015) based on Igor Pro (version 6.37). The detailed procedures were
described in Ng et al. (2011) and Sun et al. (2012). Composition dependent CE value consistent with
Sun et al. (2013) was chosen considering that aerosol samples was dried before entering the ACSM
instrument. RIEs of 5.15 and 0.7 from calibration results during the campaign using 300 nm pure
ammonium nitrate (AN) and ammonium sulfate (AS) were used for ammonium and sulfate
quantifications, while default RIEs of 1.4, 1.1 and 1.3 for organic aerosol, nitrate and chloride were



adopted. Positive matrix factorization technique with the multilinear engine (ME-2 (Canonaco et al.,
2013;Canonaco et al., 2021)) were used for resolving potential OA factors related with different
sources and processes. Four factors were deconvolved, including two primary OA factors and two
oxygenated OA factors which were usually treated as SOA. A hydrocarbon-like OA (HOA, O/C~0.15),
a cooking-like OA (COA, O/C~0.13), a less oxidized oxygenated OA (LOOA, O/C~0.7), and a more
oxidized oxygenated OA (MOOA, O/C~1.17). The mass spectra of these factors (Fig.S9) and more
details about the factor analysis could be found in Sect.S2 of the supplement.
**2.4 Organic aerosol hygroscopicity derivation**.
Organic aerosol was usually treated as nearly hydrophobic in many previous studies when
considering environmental effects of organic aerosol (Cheng et al., 2016), however quantified in this
study based on the most recently developed organic aerosol hygroscopicity quantification method by
Kuang et al. (2019). On the basis of field measurements, organic aerosol hygroscopicity parameter $\kappa$
($\kappa_{OA}$) can only be estimated through closure between directly measured overall aerosol hygroscopicity
and aerosol chemical compositions using the volume mixing rule (Kuang et al., 2020c). Kuang et al.
(2020b) developed an optical method to calculate $\kappa_{OA}$ based on the combination of the humidified
nephelometer system measurements and bulk $PM_1$ aerosol chemical-composition measurements, and
the application of this method was further manifested and discussed in Kuang et al. (2021b), thus used
in this study to estimate $\kappa_{OA}$. The humidified nephelometer system provides direct measurements of
the optical hygroscopicity parameter $\kappa_{sca}$ and aerosol scattering Ångström exponent, which can be
used together to derive a $\kappa$ value referred to as $\kappa_{f(\mathrm{RH})}$ (Kuang et al., 2017) which can be treated as the
overall aerosol hygroscopicity parameter in the hygroscopicity closure (Kuang et al., 2021a). In the
closure, ions were paired using the scheme proposed by Gysel et al. (2007) as listed in Tab.S1. Same
with Kuang et al. (2021b), κ values of ammonium sulfate (AS) and ammonium nitrate (AN) at 80%
RH were predicted using the Extended Aerosol Inorganic Model (E-AIM), and those of ammonium
chloride (AC) and ammonium bisulfate (ABS) were consistent with Liu et al. (2014). Then, the $\kappa_{OA}$
can be estimated using the following formula by assuming volume additivity and zero κ of BC:
$$\kappa_{OA} = \frac{\kappa_{f(\mathrm{RH})} - (\kappa_{AS}\varepsilon_{AS} + \kappa_{AN}\varepsilon_{AN} + \kappa_{ABS}\varepsilon_{ABS} + \kappa_{AC}\varepsilon_{AC} + \kappa_X\varepsilon_X)}{\varepsilon_{OA}} \quad (1)$$
Where ε represents volume fraction whose calculation needs total aerosol volume concentrations and





subscript represents name of an aerosol component. The total PM$_1$ aerosol volume concentration was
calculated using measurements of the dry nephelometer following the machine learning method
proposed by Kuang et al. (2018). Organic aerosol density varies over a wide range, and previous
studies demonstrate that it is tightly associated with aerosol oxidation state (Kuwata et al., 2012) and
higher O/C ratio usually corresponds to higher aerosol density. Following Wu et al. (2016) , densities
of primary organic aerosol components (POA) including HOA and COA was assumed as 1 g/cm$^3$, and
density of MOOA was assumed as 1.4 g/cm$^3$ due to its highly oxygenated feature with O/C of 1.17,
however, the density of LOOA was assumed as 1.2 g/cm$^3$ due to its moderate O/C value of 0.7. In
addition, the difference between predicted volume concentration from nephelometer measurements
and the total aerosol volume concentrations summed from known aerosol components was attributed
to aerosol components unidentified by the mass spectrometer, thus its volume fraction and
hygroscopicity were named as $\varepsilon_X$ and $\kappa_X$ in Eq.1, where $\kappa_X$ was assumed as 0.05 since the unidentified
part are usually metals and dust with low aerosol hygroscopicity. For more comprehensive discussions
on $\kappa_{OA}$ estimations as well as $\kappa_X$ treatment please refer to Kuang et al. (2021b).
**Table 1.** Densities (ρ) and hygroscopicity parameters (κ) of inorganic salts

| Species | $NH_4NO_3$ (AN) | $NH_4HSO_4$ (ABS) | $(NH_4)_2SO_4$ (AS) | $NH_4Cl$ (AC) |
|---|---|---|---|---|
| ρ (g $cm^{-3}$) | 1.72 | 1.78 | 1.769 | 1.528 |
| κ | 0.56 | 0.56 | 0.56 | 0.93 |


**3 Results and discussions**
**3.1 Overview of the pollution characteristics during Guangzhou winter.**
The timeseries of corrected aerosol scattering coefficients in dry state ($\sigma_{sp,525}$), non-refractory
PM$_1$ (NR$_{PM1}$) mass concentrations, resolved organic aerosol factors as well as meteorological
parameters are shown in Fig.1. Scattering coefficient at 525 nm of aerosols in dry state ($\sigma_{sp,525}$) varied
over a wide range of 8 to 688 Mm$^{-1}$ with an average of 118 Mm$^{-1}$. The average NR$_{PM1}$ is 20 μg/m$^3$ with
the highest NR$_{PM1}$ mass concentrations reached beyond 160 μg/m$^3$. This suggests for relatively clean
conditions compared to aerosol pollution in other polluted regions in China (Zhou et al., 2020),



however, severe pollution episodes occurred occasionally. Three haze pollution episodes characterized
by relatively high aerosol mass loading and scattering coefficients were observed before February
(Gray shaded areas in Fig.1). The evolution and formation of these three episodes were tightly
associated with the stagnant meteorological conditions as indicated by the low wind speed (< 2 m/s)
and increasing RH during the last two severe pollution episodes. As shown in Fig.1c, organic aerosol

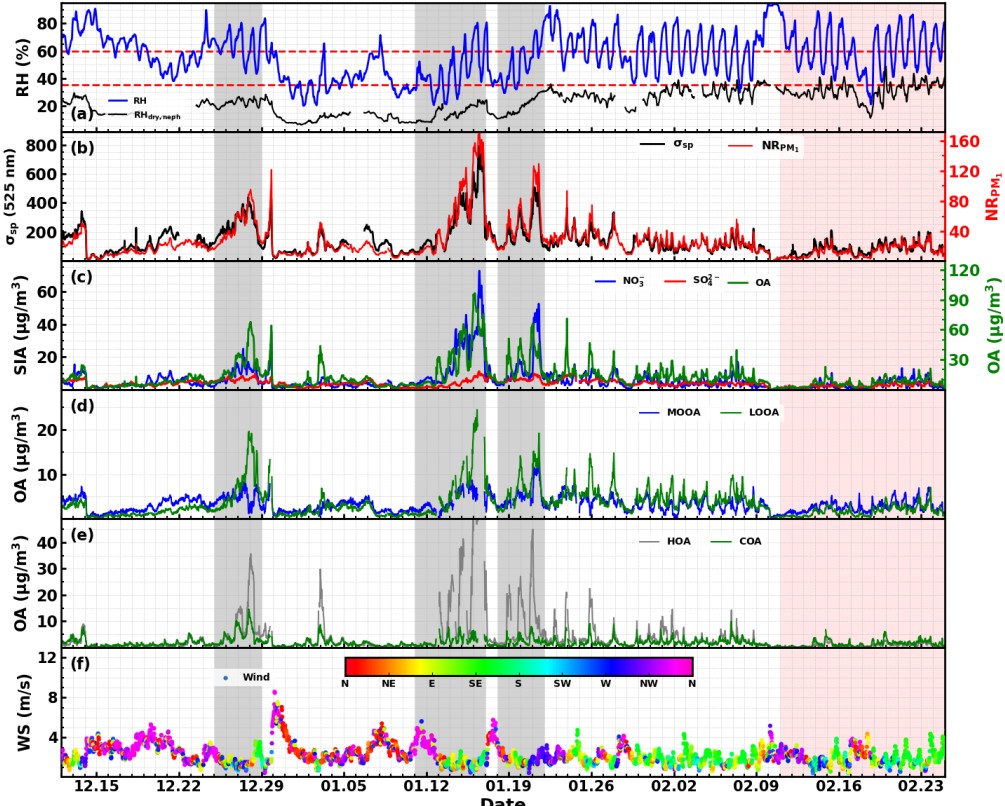

**Figure 1**. Timeseries of **(a)** RH; **(b)** aerosol scattering coefficient in dry state, and $NR_{PM1}$ concentration; **(c)** sulfate, nitrate and organic aerosol in rigt axis, **(d)** MOOA and LOOA; **(e)** HOA and COA; **(f)** wind speed and wind direction. Gray shaded areas are three identified pollution episodes, and pink shaded area is period of spring festival.

and nitrate contributed dominantly to the increase of aerosol mass, while sulfate remains almost flat
or increased slightly during these pollution episodes. For example, for the episode with the worst
pollution condition occur, the daily average $NR_{PM1}$ increased from 19 to 143 $\mu g/m^3$ from 12[th] to 16[th]
February with the organic aerosol increased from 9.3 to 69.8 $\mu g/m^3$ and nitrate increased from 5.5 to
44.2 $\mu g/m^3$, however, sulfate only increased from 1.4 to 8.5 $\mu g/m^3$. This phenomenon is quite different
from the results reported by Guo et al. (2020) and Chen et al. (2021b) that organic aerosol dominated


the aerosol mass increase with obvious increase of both sulfate and nitrate in pollution episodes in
autumn of Guangzhou urban area, however, was generally consistent with the increasing
characteristics reported by Chen et al. (2021a). These observations suggest that the aerosol pollution
differs much among seasons and years due to the highly variable characteristics of meteorological
conditions. As for the organic aerosol mass increase, the time series of resolved organic aerosol factors
are also shown in Fig.1d and Fig.1e. For the three observed pollution episodes, both increases of
primary and secondary organic aerosol (represented by summation of MOOA and LOOA) were
observed, with LOOA contributing dominantly to SOA increase and HOA contributing dominantly to
POA increase. However, the accumulation of POA (summation of HOA and COA) contributed almost
twice as much as the increase of SOA, suggesting primary emissions especially vehicle emissions
played significant roles in aerosol mass increase during pollution episodes of Guangzhou winter.
**3.2 Strong scattering ability of MOOA in dry state.**

Traditional multiple linear regression models were usually applied to determine MSEs of different

aerosol components using simultaneously measured aerosol scattering coefficients and mass
concentrations of aerosol components (Hand and Malm, 2007;Han et al., 2015;Chan et al., 1999).
However, the traditional model failed in this study due to co-variations of input variables and impactor
inconsistencies ($PM^{2.5}$ versus $PM_1$) between aerosol chemical compositions and aerosol scattering
measurements. Details about this failure was discussed in Sect.S4 of the supplement. A new
methodology was proposed to lower correlations between variables and reduce the impacts of
measurement inconsistency of aerosol populations between nephelometer and the mass spectrometer.
This method considers mainly the responses of aerosol scattering coefficient to quick mass
concentration increases of aerosol components. Using AN as an example, obvious increasing cases of
AN were identified, average changes of aerosol components as well as $\sigma_{sp,525}$ for these cases are
shown in Fig.2a. On average, AN dominated the aerosol mass increase (>90%) in these cases, however,
changes of other aerosol components differed much among cases as indicated by large standard
deviations. The $MSE_{AN}$ can be roughly estimated as around 7 $m^2/g$ if assuming $\sigma_{sp,525}$ was solely
contributed by AN increase. As shown in Fig.1d and Fig.1e, prominent increase of HOA, COA and
LOOA were frequently observed. Average changes of aerosol components and $\sigma_{sp,525}$ for identified
cases of HOA increase or COA increase are shown in Fig.2b. It shows that increases of HOA or COA
were usually accompanied with obvious increases of BC and LOOA, thus the impacts of HOA or COA





increases on observed aerosol scattering increases cannot be isolated. Similar results were obtained
with LOOA and MOOA. As shown in Fig.1, remarkable increases of LOOA cases were almost always

Figure 2. Average changes of aerosol components and $\sigma_{sp,525}$ (right axis) for identified increase cases of (a) Nitrate; (b) HOA; (c) LOOA; (d) MOOA, black error bars represent standard deviations. (e) Comparisons between observed $\sigma_{sp,525}$ changes for identified cases and multiple linear fitted values. (f) The comparison between observed $\sigma_{sp,525}$ and calculated $\sigma_{sp,525}$ using retrieved MSEs of aerosol components. Red dashed lines represent 20% deviation lines.

accompanied with the spontaneously quick HOA increase because most LOOA rapid increase cases





happened during pollution episodes and start near dusk when accumulation of vehicle emissions and
nitrate formation occurred. Thus, the average increase of LOOA was even smaller than those of AN
and HOA. Slight but obvious MOOA increase cases were also identified, and average results are also
shown in Fig.2d, showing that MOOA increase were usually accompanied with obvious nitrate
formation. These results demonstrate that MSEs of aerosol components cannot be quantified directly
from responses of aerosol scattering to aerosol emission or formation processes. However, for these
cases, mass increases of aerosol components and corresponding changes in aerosol scattering matter
most and impacts of unidentified aerosol components are reduced substantially through differential
considering the average time change for these identified cases are only 4 hours. In addition, as listed
in Tab.S3, the correlations between changes of most variables for all identified cases are much smaller
than their timeseries correlations shown in Tab.S1. Thus, the modified multiple linear regression model
$\Delta\sigma_{sp,525} = \Delta HOA \times MSE_{HOA} + \Delta COA \times MSE_{COA} + \Delta LOOA \times MSE_{LOOA} + \Delta MOOA \times MSE_{MOOA} +$
$\Delta AS \times MSE_{AS} + \Delta AN \times MSE_{AN} + \Delta BC \times MSE_{BC}$ was applied to retrieve MSEs of aerosol
components. The derived MSEs for HOA, COA, LOOA, MOOA, AN, AS and BC were 2.1, 3.9, 3.4,
9.9, 7.1, 5.5 and 3.3 m²/g, respectively. The fitted $\Delta\sigma_{sp,525}$ correlated highly (R²=0.98, average ratio
1.0) with observed $\Delta\sigma_{sp,525}$ as shown in Fig.2e. Derived MSEs were used to calculate $\sigma_{sp,525}$ during
the whole observation period using the formula $\sigma_{sp,525} = 2.1 \times HOA + 3.9 \times COA + 3.4 \times LOOA +$
$9.9 \times MOOA + 5.5 \times AS + 7.1 \times AN + 3.3 \times BC$ , and compared with observed $\sigma_{sp,525}$ . Good
consistency (R²=0.93 and average ratio of 1.05, Fig.2f) was achieved between calculated and observed
$\sigma_{sp,525}$ values. In addition, the retrieved MSE_AN (7.1) using the modified multilinear regression model
was quite consistent with the estimated one (7 m²/g) based on average changes shown in Fig.2a, which
indirectly confirms the reliability of the modified method.

Tao et al. (2019) quantified MSEs of fine mode AS, AN as well as elemental carbon (EC) using

size-resolved filter measurements in four seasons of Guangzhou urban area. Their results demonstrate
that MSEs of AN and AS bears relatively large standard deviations and variations among seasons,
however, MSE of EC varied little among seasons with small standard deviations (2.6±0.1 m²/g). The
derived MSE_BC of 3.3 m²/g was close to the MSE_EC reported in Tao et al. (2019). The derived MSE_AN
and MSE_AS were obviously higher than those reported in previous studies in which MSE_AN and MSE_AS
were estimated through Mie theory of size-resolved filter measurements [Tao et al., 2019;Chen et al., 2020]. For
example, Tao et al. (2019) reported MSEs of 4.4±1.3 m²/g for AN and 4.3±0.9 m²/g for AS in winter





of urban Guangzhou for fine mode aerosols (<2.1 $\mu$m). The reason explaining this inconsistency is that
the derived MSE$_{AN}$ using multiple regression method here is $MSE^*_{AN} = \frac{\sigma_{sp,525}(PM_{2.5})}{[AN](PM_1)}$, however, MSE$_{AN}$
derived for example in Tao et al. (2019) of fine mode is $MSE_{AN,PM_{2.1}} = \frac{\sigma_{sp,525}(PM_{2.1})}{[AN](PM_{2.1})}$. The MSE$_{AN,PM_{2.1}}$,
MSE$_{AN,PM_1}$ and MSE$^*_{AN}$ values of 4.4, 5.3 and 7.5 m$^2$/g are simulated using the reported average AN
mass size distributions reconstructed from size-resolved filter measurements in winter of urban
Guangzhou by Tao et al. (2019) (as shown in Fig.3a) as inputs of Mie model. The simulated MSE$^*_{AN}$
of 7.5 is very close to the retrieved MSE$^*_{AN}$ and is much higher than simulated MSE$_{AN,PM_1}$ due to
substantial mass contributions of 1 to 2.1 μm as shown in Fig.3a, demonstrating that good consistency
between results of the multiple regression model and results of Tao et al. (2019) was achieved.

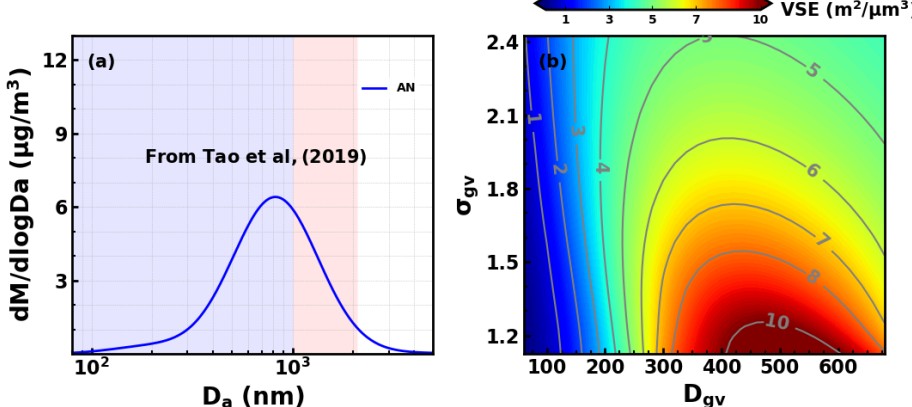

**Figure 3. (a)** AN mass size distributions derived by Tao et al, (2019) based on size-resolved filter measurements, Da is the aerodynamic diameter; (b) Simulated aerosol volume scattering efficiency (VSE) under different volume size distributions through varying volume geometric mean D$_{gv}$ and standard deviation $\sigma_{gv}$ of lognormal size distributions. Blue shaded area corresponding to PM$_1$ and pink shaded area corresponding to PM$_{1-2.1}$.

If using the simulated ratio MSE$_{AN,PM_1}$/MSE$^*_{AN}$ to approximate MSE$_{AN,PM_1}$, MSE$_{AS,PM_1}$ and

MSE$_{MOOA,PM_1}$, values of 5.0, 3.9 and 6.9 m$^2$/g for MSE$_{AN,PM_1}$, MSE$_{AS,PM_1}$ and MSE$_{MOOA,PM_1}$ would
be obtained, which falls in the reported ranges of MSE$_{AS}$ and MSE$_{AN}$ (Tao et al., 2019), however, the
high MSE$_{MOOA,PM_1}$ needs further investigation. MSE is determined by the aerosol volume scattering
efficiency (VSE) defined as aerosol scattering per unit aerosol volume and aerosol density ($\rho_a$) with
$MSE = \frac{VSE}{\rho_a}$. High MSE$^*_{MOOA}$ of 9.9 m$^2$/g was retrieved, however, most of the difference between
MSE$^*_{MOOA}$ and MSE$^*_{AN}$ might be explained by their density differences. Based on the Mie theory,
aerosol scattering is in general proportional to aerosol volume (Kuang et al., 2018), thus the volume



size distribution are determining factors in VSE variations. The VSE of $PM_1$ under different unimodal
volume lognormal distribution conditions with refractive index of $1.53\text{-}10^{-7}i$ were simulated and shown
in Fig.3b. The approximated $MSE_{AN,PM_1}$ and $MSE_{MOOA,PM_1}$ of 5.0 and 6.9 $m^2/g$ corresponds to
$VSE_{AN,PM_1}$ and $VSE_{MOOA,PM_1}$ of 8.6 and 9.7 $m^2/\mu m^3$ according to the aerosol densities discussed in
Sect. 2.4, falling within the VSE ranges of geometric mean diameter ($D_{gv}$) near 500 nm and geometric
standard deviation ($\sigma_{gv}$) of 1.3 to 1.5. This result is consistent with conclusions of several previous
studies that the MOOA with highest oxygen state that have experienced complex chemical aging such
as aqueous phase reactions likely share similar shape of mass/volume size distribution with inorganic
secondary aerosols (Kuang et al., 2020a;Wang et al., 2021), and this result rationalizes that using
$MSE_{AN,PM_1}/MSE_{AN}^*$ ratio to derive $MSE_{AS,PM_1}$ and $MSE_{MOOA,PM_1}$. In addition, aerosol refractive index
also played significant roles in aerosol VSE variations (Zhao et al., 2019b;Liu et al., 2013), and the
high MSE of MOOA might also be related with the high real part of its refractive index. Laboratory
results of Li et al. (2017) revealed enhanced light scattering of SOA formed through multiphase
reactions due to increase of the real part of the refractive index. Zhao et al. (2021) reported that real
part of aerosol refractive index varied over a wide range (1.36 to 1.78), and in general increased with
the mass fraction increase of organic aerosol, suggesting generally higher real part of refractive index
of organic aerosol. In general, these results revealed strong scattering abilities of MOOA under dry
state, however the size distribution and refractive index of MOOA needs further comprehensive
investigations.

Moreover, effective densities of HOA, COA, LOOA are near 1 $g/cm^3$, suggesting that VSEs of

HOA, COA, LOOA are around or slightly higher than their corresponding MSEs. As shown in Fig.3b,
$VSE_{PM_1} < 4$ $m^2/\mu m^3$ means that $D_{gv}$ was generally lower than 250 nm, thus more than 99% of aerosol
mass resided in $PM_1$ under $\sigma_{gv} < 1.8$. Therefore, derived MSEs of HOA, COA, LOOA can be treated
as their corresponding $MSE_{PM_1}$ values. Cai et al. (2020) reported average HOA and COA volume size
distributions in urban Beijing using PMF techniques. They found (Fig.7 in Cai et al. (2020)) that HOA
volume peaked near 200 nm, and COA volume size distribution showed bimodal characteristics with
the first mode peaking near 90 nm and the second mode peaking near 350 nm, yielding MSEs that
share similar magnitude with the retrieved MSEs of HOA and COA. These results further confirmed
the reliability of the newly proposed multiple regression method.





### 3.3 Water uptake abilities of organic aerosols


Timeseries of derived $\kappa_{OA}$ is shown in Fig.4a, estimated $\kappa_{OA}$ ranged from -0.08 to near 0.35 with
an average of 0.09 which is in general consistent with those reported in other regions (Kuang et al.,
2020c). It was found that variations of derived $\kappa_{OA}$ correlated tightly with mass fractions of MOOA
(R=0.6) and POA (R=-0.52) as shown in Fig.4b and Fig.4c. The MOOA enhanced the overall $\kappa_{OA}$ and
POA lowered $\kappa_{OA}$, which is consistent with conclusions of previous studies, however, no correlations
were found between $\kappa_{OA}$ and mass fractions of LOOA (R=0.06). As shown in Fig.4d and Fig.4e,
drastic increase of POA before dusk would bring drastic decrease of $\kappa_{OA}$ to around 0.05, which are in
accordance with reported results in previous literature that most POA components are hydrophobic
with $\kappa$ of almost zero. Assuming $\kappa$ values of HOA and COA as zero, the multilinear formula
considering ZSR mixing rule with $\kappa_{OA} = \kappa_{MOOA} \times \varepsilon_{MOOA} + \kappa_{LOOA} \times \varepsilon_{LOOA}$ was used to fit the average
diurnal variations of $\kappa_{OA}$ with average volume fractions of MOOA and LOOA ($\varepsilon_{MOOA}$ and $\varepsilon_{LOOA}$) in
total organic aerosol as inputs. The fitted results are shown in Fig.4e ($R^2$=0.89, average ratio of 1),
yielding $\kappa_{MOOA}$ and $\kappa_{LOOA}$ values of 0.23 and 0.13. The relatively lower value of $\kappa_{LOOA}$ and
sometimes co-increase phenomena of LOOA and POA during dusk period (as shown in Fig.1 and
Fig.3d) explain the weak correlations between $\kappa_{OA}$ and LOOA mass fraction. Most previous field-
based organic aerosol hygroscopicity studies focused only on the overall $\kappa_{OA}$ characterization of entire
organic aerosol population, and rarely specific to secondary organic aerosol factors (Kuang et al.,
2020c). Considering the estimated O/C ratios of 1.17 and 0.7 for MOOA and LOOA, the retrieved
$\kappa_{LOOA}$ and $\kappa_{MOOA}$ values are consistent with previous findings that organic aerosol oxidation state
impacts significantly on organic aerosol hygroscopicity and usually higher hygroscopicity of more
oxygenated organic aerosol (Kuang et al., 2020c). Especially, Lambe et al. (2011) investigated the
relationships between organic aerosol hygroscopicity of laboratory generated SOA of varying kinds of
volatile organic compounds using cloud condensation nuclei activity measurements and reported a
linear relationship of $\kappa_{OA} = (0.18 \pm 0.04) \times O/C + 0.03$, yielding $\kappa_{OA}$ values of 0.24 and 0.16 for
O/C of 1.17 and 0.7, which is slight higher but generally consistent with retrieved ones of this study.
In addition, the retrieved $\kappa_{LOOA}$ and $\kappa_{MOOA}$ values fall well within the predicted relationship band
between O/C and intrinsic $\kappa_{OA}$ in Wang et al. (2019a) , implying that the hygroscopicity of SOA at RH
of 80% during this field campaign might not be limited by solubility (Liu et al., 2018). Overall, the
retrieved $\kappa_{LOOA}$ and $\kappa_{MOOA}$ are further verified indirectly through comparison with previous literature



results, confirming the strongest water uptake abilities of MOOA among OA factors.

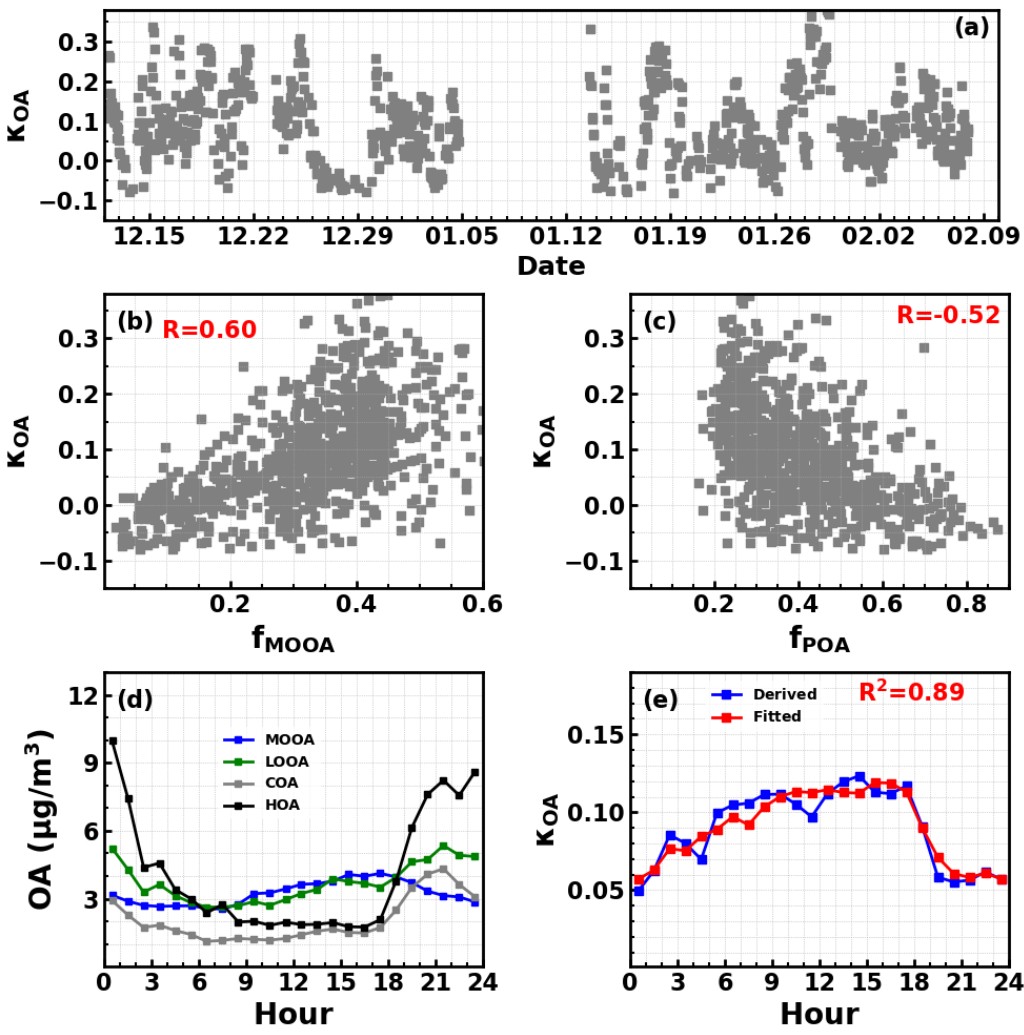

**Figure 4**. Timeseries of derived $\kappa_{OA}$ (a); Correlations between derived $\kappa_{OA}$ and mass fractions of MOOA (b), POA (c) in total organic aerosols. (d) Average diurnal variations of derived $\kappa_{OA}$ (blue) and corresponding fitted ones. (e) Average diurnal variations of resolved mass concentrations of organic aerosol factors.


**3.4 Dominant contribution of MOOA to organic aerosol scattering ability**.

High scattering efficiency and water uptake abilities of MOOA resulted in the strongest light
scattering abilities of MOOA among all organic aerosol factors. As shown in Fig.5a and Fig.5b, on
average, 34% mass contribution of MOOA to entire $PM_1$ organic aerosol populations, however,





contributed 51% of organic aerosol scattering in dry state. The dominant role of MOOA in organic
aerosol scattering would be further enhanced under ambient RH conditions due to the highest water
uptake abilities of MOOA among organic aerosol factors. Results of Kuang et al. (2017) demonstrate
that hygroscopicity parameter $\kappa$ can be directly linked to optical hygroscopicity parameter $\kappa_{sca}$
through $\kappa_{sca} = \kappa \times R_{sca}$, and thus aerosol light scattering enhancement factor f(RH). Particle number
size distributions plays the most important role in $R_{sca}$ variations with $\kappa$ plays the smaller role. Yu et
al. (2018) investigated $R_{sca}$ variations from measurements of several field campaigns, found that $R_{sca}$
varied within the range of 0.55 to 0.8 with an average of 0.66 and parameterized $R_{sca}$ with scattering
Ångström exponent. Here, the relationship between $VSE_{PM1}$ and $R_{sca}$ were further simulated using
Mie theory through varying volume geometric mean $D_{gv}$ of lognormal size distributions from 100 to
700 nm considering that aerosol size distributions also play the dominant role in $VSE_{PM1}$ variations.
Simulated results are shown in Fig.S11, demonstrating that $R_{sca}$ decreased almost linearly with the
increase of $VSE_{PM1}$ for $VSE_{PM1} < \sim 6 \ m^2/\mu m^3$, however, varied complexly with $VSE_{PM1}$ for
$VSE_{PM1} > \sim 6 \ m^2/\mu m^3$. According to estimated $MSE_{PM1}$ values of LOOA, MOOA, AS, AN in Sect 4.2
and their densities introduced in Sect.2.4. $VSE_{PM1}$ values of LOOA, MOOA, AS and AN are 4.08, 9.6,
6.9 and 8.7 m²/μm³, respectively. Accordingly, the $R_{sca,LOOA}$ was estimated as 0.87, and 0.63 as an
average estimate was used for $R_{sca,MOOA}$, $R_{sca,AS}$ and $R_{sca,AN}$. MSEs of LOOA, MOOA, AS and AN
with their water uptake abilities considered under different RH conditions can thus be estimated using
$MSE_{PM1,X}(RH) = MSE_{PM1,X,dry} \times (1 + \kappa_X \times R_{sca,X} \times \frac{RH}{100-RH})$.
The problem remain that if continuous increase of MSE as a function of ambient RH increasing
can be applied, because aerosol phase states were also crucial in determining the responses of aerosol
scattering to RH increases (Kuang et al., 2016). Many factors such as ambient RH (Liu et al., 2017b;Liu
et al., 2016), RH history aerosol particles have been experienced (Kuang et al., 2016), deliquescent
and crystalline properties determined by aerosol chemical compositions and mixing states are
important in determining aerosol phase state (Li et al., 2021). The ambient RH ranged from 20 to 94
with an average of 57%, with the histogram of ambient RH also shown in Fig.S12. Results of Liu et
al. (2017b) found a transition from semisolid to liquid state at RH of about 60%, suggesting ambient
aerosol particle might exist as semisolid or solid phase at low RH conditions. The phase state of
ambient aerosols depends not only on ambient RH but also the RH history that aerosols have



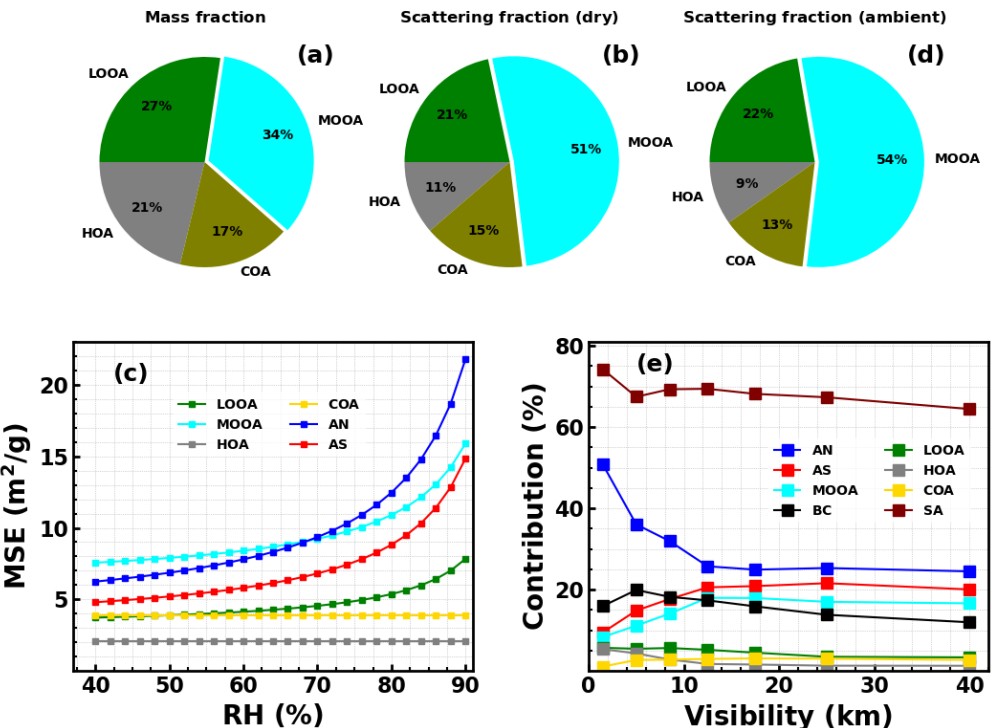

Figure 5. (a) Average mass fractions of organic aerosol components in total organic aerosol; Contributions of different organic aerosol components to total organic aerosol scattering coefficients under dry (b) and ambient (d) conditions for PM$_1$. (c) MSEs of aerosol components under different RH conditions. (e) Contributions of different aerosol components to visibility degradation under different visibility conditions.

experienced (Kuang et al., 2016). For instance, aerosols under ambient RH of 40% in the afternoon
would remain liquid if their crystalline RH is lower than 40% and they experienced high RH conditions
(such as >80%) during the morning. Therefore, the lowest RH that aerosols have experienced in the
afternoon and highest RH they have experienced in the morning are crucial for their phase state for
hydrophilic aerosols. Besides, the deliquescent RH and crystalline RH are another two crucial
parameters, however, quite complex for ambient multicomponent aerosols (Kuang et al., 2016;Li et
al., 2021). Scanning RH of 60-90% was set-up for the humidified nephelometer system from 13[th] to
26[th] January and continuous increase of aerosol light scattering enhancement factor were always
observed (Fig.S13) with the RH in the dry nephelometer are always lower than 35% (Fig.1a),
suggesting that aerosols were either not crystalized under RH of <35% or were deliquesced under RH
of < 60%. The lines of 35% and 60% are plotted in Fig.1a and most of days (>85%) were either with





its lowest RH >35% or with its highest RH>60%, suggesting liquid state in most times for internally
mixed ambient secondary aerosols, and continuous increase of aerosol light scattering as ambient RH
changes.

Therefore, continuous increases of MSEs of LOOA, MOOA, AN and AS as a function of RH

were considered, and results are shown in Fig.5c. $MSE_{PM1}$ of MOOA changed from 6.9 $m^2/g$ under
dry conditions to 11 and 16 $m^2/g$ (corresponding to f(RH) values of 1.6 and 2.3) under RH of 80% and
90%, revealing that scattering abilities of MOOA would be largely enhanced by aerosol hygroscopic
growth. The $MSE_{PM1}$ of LOOA was enhanced from 3.4 under dry condition to 7 $m^2/g$ of 90% RH,
however, $MSE_{PM1}$ values of HOA and COA remained constant due to their hydrophobic properties.
Both $MSE_{PM1}$ values of AS and AN increased quickly as a function of RH, and their f(RH) values
reached as high as 2.5 and 2.2 at RH of 80%. The $MSE_{PM1}$ of AN exceeded that of MOOA at RH near
70%, however, $MSE_{PM1}$ of MOOA was always higher than that of AS for RH<90%.  Average $MSE_{PM1}$
values of secondary aerosol components considering water uptake under ambient RH conditions
during the observation period are 6.8, 8.5, 8.9 and 4.2 $m^2/g$ for AS, AN, MOOA and LOOA,
respectively. Demonstrating strongest scattering abilities of MOOA under ambient air conditions
during the observation period. Thus, the average contribution of MOOA to $PM_1$ organic aerosol
scattering are further enhanced to 54% under ambient conditions as shown in Fig.5d (campaign
average RH ~57%).

**4. Implications for visibility improvement and aerosol radiative effects simulations**.

The strong light scattering abilities of MOOA might have significant effects on atmospheric

visibility and direct aerosol radiative effects, thus have broad implications for both aerosol
environmental and climate effects. The contributions of MOOA to visibility degradation under
different visibility conditions were estimated as MOOA contributions to ambient atmospheric
extinction caused by both aerosols and air molecules. The results are shown in Fig.5e, and detailed
estimation method is introduced in Sect.S6 of the supplement. It shows that AN contributed most to
visibility degradation especially under polluted conditions, which is consistent with findings of several
recent studies that nitrate plays increasing and event dominant role in visibility degradation (Liu et al.,
2020;Li et al., 2022) in several regions of China. MOOA contributed slightly smaller than ammonium
sulfate and their contributions have increased to ~20% for visibility ranges of 10-20 km. Contribution


of MOOA to total organic aerosol scattering under ambient conditions was slightly higher than that
under dry conditions due to the relatively low RH conditions during winter. However, the contributions
of MOOA might be much higher during other seasons due to much higher RH conditions, for example,
yearly average RH of >70% in Guangzhou (Xu et al., 2020). Overall, secondary aerosols contributed
dominantly to visibility degradation (~70% on average), and MOOA represented the third contributor
among secondary aerosol components (16% on average), demonstrating significant impacts of MOOA
to visibility degradations. Thus, more attentions should be paid to property changes of SOA regarding
visibility improvement investigations and policy making. Moreover, MOOA with high scattering
abilities would likely contribute substantially to aerosol optical depth, and the accurate estimations of
organic aerosol radiative effects in models need accurate representations of MOOA even its mass
contribution to organic aerosol is small. However, constant scattering efficiency for organic aerosols
derived from fixed size distributions and refractive index were usually assumed in current chemical
models (Latimer and Martin, 2019) and unsatisfactory performance of current models in SOA
simulations were generally reported, which hinders the accurate representations of direct aerosol
radiative effects.

The role of MOOA would be likely getting more important in future due to stringent control

policy of precursors of inorganic aerosol components such as sulfur dioxide and nitrogen oxides. Both
formation pathways and precursor sources of MOOA are complex, however, current understandings
about physical and chemical properties as well as formation pathways of MOOA remain limited.
Therefore, call for more comprehensive studies on formation, evolution, and physical properties of
MOOA, to better parameterize optical properties of MOOA in models and implement targeted control
strategies on MOOA precursors of volatile organic compounds in the future.

**Supporting information**
Supporting information on site information, Q-ACSM PMF analysis, traditional multiple linear
regression model analysis and visibility contribution estimation method, including 2 supporting tables
and 13 supporting figures.

**Competing interests**. The authors declare that they have no conflict of interest.





**Author Contributions**. YK and LL designed the aerosol experiments. YK conceived and led this research. LL and YK wrote the manuscript. MMZ and LL conducted the Q-ACSM measurements. MMZ and YH performed the PMF analysis. BX and YK performed the humidified nephelometer system measurements. All other coauthors have contributed to this paper in different ways.

## Acknowledgments

This work is supported by the National Key Research and Development Program of China (2019YFCO214605 and 2016YFC0202000); National Natural Science Foundation of China (41805109 and 42105092); Guangdong Basic and Applied Basic Research Foundation (2019A1515110791) ; Science and Technology Innovation Team Plan of Guangdong Meteorological Bureau (GRMCTD202003); Science and Technology Research Project of Guangdong Meteorological Bureau (GRMC2018M07)  and Natural Science Foundation of Fujian Province, China (2021J01463).

**Data availability**. The data used in this study are available from the first author and corresponding author upon request.

Li Liu (liul@gd121.cn) and Ye Kuang (kuangye@jnu.edu.cn)

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
