# Peer review of "Strong light scattering of highly oxygenated organic aerosols impacts significantly on"

_Atmospheric Chemistry and Physics, 2022_

## Author Response (AR1)

**Dear Editor:**

Thanks for your time. We are grateful for the reviewer's careful inspection of our manuscript. All these comments raised by the referees have been explicitly replied point by point and incorporated into the revision.

Thank you very much for your attention and consideration.

Sincerely Yours

Ye Kuang

**Responses to anonymous referee #1**

**General Comment:**

This work tried to estimate the relative importance of secondary organic aerosol in visibility impairment via light scattering. Based on the field measurements on aerosol components and aerosol extinction in Guangzhou city, the authors found that more oxidized oxygenated organic aerosol is the most efficient light scattering aerosol component due mainly to its large mass proportion and high hygroscopicity, which highlights the importance of light extinction of organic aerosol in visibility degradation. Generally, this manuscript is well written and fits the scope of ACP. I have a few comments as listed below.

**Response**: Many thanks, we have improved the manuscript according to your comments.

**Major comments:**

**Comment**: The principal finding of the work is the hygroscopicity and light extinction of secondary organic aerosol. I do think that the method for estimating hygroscopicity and light extinction of various secondary organic aerosols as well as its validation need to be detailed. I am a little bit confused about the treatment of organic aerosol hygroscopicity parameters for different organic components, especially for MOOA and LOOA. More descriptions and arguments are suggested to be added in Section 2.

**Response**: Many thanks for your suggestion. More details about  $\kappa_{MOOA}$  and  $\varepsilon_{LOOA}$  derivations are added in Sect.2.4 of the revised manuscript as the following:

"Using the ZSR mixing rule, the  $\kappa_{OA}$  derived at RH of 80% can be further expressed as:

 $\kappa_{OA} = \varepsilon_{HOA} \times \kappa_{HOA} + \varepsilon_{COA} \times \kappa_{COA} + \varepsilon_{MOOA} \times \kappa_{MOOA} + \varepsilon_{LOOA} \times \kappa_{LOOA}$ (2)

Where  $\varepsilon$  represents volume fractions of primary and secondary organic aerosol components in total organic aerosols. Assuming  $\kappa$  values of HOA and COA as zero, Eq.2 can be simplified as  $\kappa_{OA} = \kappa_{MOOA} \times \varepsilon_{MOOA} + \kappa_{LOOA} \times \varepsilon_{LOOA}$ . Considering the noisy characteristics of derived  $\kappa_{OA}$  as shown in Fig.4a, this simplified formula was not directly used to fit all derived  $\kappa_{OA}$  values. Instead, average diurnal variations of derived  $\kappa_{OA}$  were firstly acquired and then fitted using  $\kappa_{OA} = \kappa_{MOOA} \times \varepsilon_{MOOA} + \kappa_{LOOA} \times \varepsilon_{LOOA}$  with average diurnal volume fractions of MOOA and LOOA ( $\varepsilon_{MOOA}$  and  $\varepsilon_{LOOA}$ ) in total organic aerosol as inputs, which yields average  $\kappa_{MOOA}$  and  $\kappa_{LOOA}$ ."

**Minor comments**

**Comment**: Line 43-45: This sentence is suggested to be rephrased for clarity.

**Response**: Thanks, this sentence is revised as "Overall organic aerosol hygroscopicity parameter  $\kappa_{OA}$  was quantified directly through hygroscopicity closure, however, hygroscopicity parameters of SOA components were further retrieved using multilinear regression model by assuming hydrophobic properties of primary organic aerosols."

**Comment**: Figure 1: typo "right axis" in the caption; SIA need to be spelt out in Figure 1c.

**Response**: Revised accordingly.

Comment: Line 230: typo "PM2.5".

**Response**: Revised accordingly.

**Responses to anonymous referee #2**

**General comment:**

Liu et al. measured the chemical composition and light scattering coefficient of wintertime organic and inorganic components of ambient aerosols in Guangzhou, China, with an aerosol chemical speciation monitor and a tandem nephelometer system. Positive matrix factorization was applied to the Q-ACSM data to extract primary and secondary organic aerosol factors. The dry mass scattering efficiency of the organic aerosol factors ranged from 2.1 m2/g (hydrocarbon-like OA) to 9.9 m2/g (MOOA), and the MSE of MOOA increased to 16 m2/g at 90% RH. A hygroscopicity parameter of 0.23 was obtained for MOOA from light scattering enhancement factors measured under humidified conditions relative to dry conditions. Overall, MOOA contributed 54% of the ambient OA scattering and 20% of the ambient non-refractory aerosol scattering.

Response: We thank the reviewer for all the valuable comments and suggestions,

**Major Comments:**

**Comment**: The authors claim that calculated  $\kappa_{OA}$  values are not solubility-limited above RH ~ 80% (e.g. L349). To support this hypothesis, I think it would be useful to add a plot showing  $\kappa_{OA}$  as a function of RH (over the range of conditions accessed in the humidified nephelometer) for the LOOA and MOOA factors. This result could then be compared to humidity-dependent  $\kappa_{OA}$  values measured in, for example, biogenic SOA that spans a range of phase state/viscosity (e.g. Pajunoja et al., 2015). Adding these details should also clarify the conditions that were used to calculate the  $\kappa_{OA}$  values that are discussed in the text – for example, it is not clear to me from the text what humidity condition(s) were used to obtain  $\kappa_{OA} = 0.23$  for MOOA.

**Response:** Thanks for the excellent suggestion. We agree with the reviewer that the discussion of solubility influences on  $\kappa_{OA}$  should include more informations about its RH dependence. The discussion of solubility influence is only a tiny piece of this paragraph, and not the focus of this manuscript at all, we have deleted this sentence. Actually, after submitting this manuscript, we realized we can dig more into the RH dependence of  $\kappa_{OA}$  and their influencing factors by using the humidified nephelometer and aerosol chemical composition measurements. And we are now preparing a manuscript about the RH dependence of  $\kappa_{OA}$  which reveals increasing trend of  $\kappa_{OA}$  as a function a RH which is similar with the hygroscopic behavior of SOA with high O/C in (Pajunoja et al., 2015), however varies a lot under varying conditions. Insightful analysis about their influencing factors would be laid down in this manuscript. We really thank you for this valuable suggestion, we believe this suggestion would improve our discussions on RH dependence of SOA hygroscopicity in the new manuscript.

To make it more clear what humidity condition(s) were used to obtain  $\kappa_{OA}$ . The  $\kappa_{OA}$  derivation part is revised as:

"Same with Kuang et al. (2021), retrieved  $\kappa_{f(RH)}$  at RH of 80% was used as measured average  $\kappa$  of PM1 aerosol populations,  $\kappa$  values of ammonium sulfate (AS) and ammonium nitrate (AN) at 80%

RH were predicted using the Extended Aerosol Inorganic Model (E-AIM), and those of ammonium chloride (AC) and ammonium bisulfate (ABS) were consistent with Liu et al. (2014). Then, the  $\kappa_{OA}$  at RH of 80% can be estimated using the following formula by assuming volume additivity and zero  $\kappa$  of BC:"

And the following part is added in Sect 2.4 to make it more clear how  $\kappa_{MOOA}$  and  $\kappa_{LOOA}$  were derived:

"Using the ZSR mixing rule, the  $\kappa_{OA}$  derived at RH of 80% can be further expressed as:

 $\kappa_{OA} = \varepsilon_{HOA} \times \kappa_{HOA} + \varepsilon_{COA} \times \kappa_{COA} + \varepsilon_{MOOA} \times \kappa_{MOOA} + \varepsilon_{LOOA} \times \kappa_{LOOA} \quad (2)$

Where  $\varepsilon$  represents volume fractions of primary and secondary organic aerosol components in total organic aerosols. Assuming  $\kappa$  values of HOA and COA as zero, Eq.2 can be simplified as  $\kappa_{OA} = \kappa_{MOOA} \times \varepsilon_{MOOA} + \kappa_{LOOA} \times \varepsilon_{LOOA}$ . Considering the noisy characteristics of derived  $\kappa_{OA}$  as shown in Fig.4a, this simplified formula was not directly used to fit all derived  $\kappa_{OA}$  values. Instead, average diurnal variations of derived  $\kappa_{OA}$  were firstly acquired and then fitted using  $\kappa_{OA} = \kappa_{MOOA} \times \varepsilon_{MOOA} + \kappa_{LOOA} \times \varepsilon_{LOOA}$  with average diurnal volume fractions of MOOA and LOOA ( $\varepsilon_{MOOA}$  and  $\varepsilon_{LOOA}$ ) in total organic aerosol as inputs, which yields average  $\kappa_{MOOA}$  and  $\kappa_{LOOA}$ ."

**Minor Comments**:**

**Comment**: L108-L109: Typo – "8-9 L" and "8 L" should presumably have units of L/min.

Response: Revised accordingly.

**Comment**: L146: Please state the range of composition-dependent ACSM collection efficiency values that were calculated.

**Response**: Thanks, this part is revised as:

"Composition dependent CE value consistent with Middlebrook et al. (2012) and Sun et al. (2013) was chosen considering that aerosol samples was dried before entering the ACSM instrument. According to Middlebrook et al. (2012), CE = max (0.45,  $0.0833+0.9167\times$ ANMF), where ANMF is the ammonium nitrate mass fraction in NR-PM1. The results showed that about 10% of samples had CE values larger than 0.45, with the largest value of 0.65. The average CE value of the samples with a CE greater than 0.45 was 0.5."

**Comment**: Figure 2: In my opinion, the information shown in panels (a)-(d) would be more useful if summarized in a table. Figures 2e and 2f might be possible to move to the supplement.

**Response**: Thanks for the suggestion. Fig.2 (a)-(d) describe well how our modified multilinear regression model was done. And readers are likely not interested in the concrete values of aerosol concentrations changes. Fig.2e and Fig.2f represents the general performance of the revised regression methods. After careful consideration, we decide to keep the current form of Fig.2.

**Comment**: **Figure 3**: Why does this figure need to be presented in the main paper? If I understand it correctly, it is a result obtained in a previous study (Tao et al., 2019), not

**in this one.**

**Response**: Yes, part of Fig.3a is obtained from (Tao et al., 2019), however, we modified the presentation form of this result for reader's readability. Actually, we also struggled about if move Fig.3a to the supplement. But we finally decide that Fig.3a is very important for the MSE discussions, put in the main paper is better for readers' convenience.

**Comment:** Figure 4: Please make the x-axis scale the same in panels (a) and (b). It would be useful to decrease the minimum  $f_{MOOA}$  and  $f_{POA}$  values to zero. Also, I would replace " $f_{POA}$ " with " $f_{HOA}$  +  $f_{COA}$ " (if that is what it represents) to directly relate it to the PMF factors.

**Response**: Many thanks for this good suggestion. Revised accordingly.

**Comment: Figure 5:**

(1) The layout of this figure is confusing: the pie charts on top are labeled as panels (a),
(b), and (d); it would make more sense to label these as panels (a), (b), and (c). **Response**: Revised accordingly.

(2) In the "MSE vs RH plot" (Fig. 5c), please show MSE = 0 on the axis scale.Response: Revised accordingly.

(3) In the "% Contribution vs Visibility (km)" plot (Fig. 5e), please show % Contribution= 0 on the plot.

Response: Revised accordingly.

(4) Fig 5e: I suggest changing the label "% Contribution" to "Fractional Aerosol Scattering " or something that more clearly explains what is being shown.Response: Thanks, the label was changed as "Extinction fraction (%)"

(5) Fig 5e: I don't understand what "Visibility (km)" means. I cannot find where this is discussed/explained in the manuscript. Please add a few sentences to the text to explain what this figure is showing.

**Response**: Thanks for the reminding. The relating part is revised as:

"Atmospheric visibility measures the maxima distance that people can see, which is determined by atmospheric extinction. The strong light scattering abilities of MOOA might have significant effects on atmospheric visibility and direct aerosol radiative effects, thus have broad implications for both aerosol environmental and climate effects. The contributions of different aerosol components to visibility degradation under different visibility conditions were estimated as fractional contributions to ambient atmospheric extinction caused by both aerosols and air molecules. The results are shown in Fig.5e, and detailed estimation method is introduced in Sect.S6 of the supplement."

(6) Fig 5e: What does the "SA" symbol mean?

**Response**: "SA corresponds to summation of secondary aerosol components" is added in the figure caption.

(7) Fig 5e: I am confused about how to interpret the numerical values shown in this figure. If the fractional scattering contribution is shown, shouldn't the values add up to 100%? If not, why? It would be easier for me to understand this plot if it was normalized so that the fractional scattering contributions summed to 100% or 1.

**Response**: This figure showed the fractional contribution of aerosol components to ambient atmospheric extinction, extinction contributions of gas Rayleigh scattering and absorption (NO2) are also accounted for visibility calculation. So, summation of extinction fractions attributed to aerosol components are not 1. The following sentence is added in the manuscript to make this clearer:

"The contributions of different aerosol components to visibility degradation under different visibility conditions were estimated as fractional contributions to ambient atmospheric extinction caused by both aerosols and air molecules."

**References**

Kuang, Y., Huang, S., Xue, B., Luo, B., Song, Q., Chen, W., Hu, W., Li, W., Zhao, P., Cai, M., Peng, Y., Qi, J., Li, T., Wang, S., Chen, D., Yue, D., Yuan, B., and Shao, M.: Contrasting effects of secondary organic aerosol formations on organic aerosol hygroscopicity, Atmos. Chem. Phys., 21, 10375-10391, 10.5194/acp-21-10375-2021, 2021.
Liu, H. J., Zhao, C. S., Nekat, B., Ma, N., Wiedensohler, A., van Pinxteren, D., Spindler, G., Müller, K., and Herrmann, H.: Aerosol hygroscopicity derived from size-segregated chemical composition and its parameterization in the North China Plain, Atmos. Chem. Phys., 14, 2525-2539, 10.5194/acp-14-2525-2014, 2014.
Pajunoja, A., Lambe, A. T., Hakala, J., Rastak, N., Cummings, M. J., Brogan, J. F., Hao, L., Paramonov, M., Hong, J., Prisle, N. L., Malila, J., Romakkaniemi, S., Lehtinen, K. E. J., Laaksonen, A., Kulmala, M., Massoli, P., Onasch, T. B.,

Donahue, N. M., Riipinen, I., Davidovits, P., Worsnop, D. R., Petäjä, T., and Virtanen, A.: Adsorptive uptake of water by semisolid secondary organic aerosols, Geophysical Research Letters, 42, 3063-3068, 10.1002/2015GL063142, 2015.

Sun, Y. L., Wang, Z. F., Fu, P. Q., Yang, T., Jiang, Q., Dong, H. B., Li, J., and Jia, J. J.: Aerosol composition, sources and processes during wintertime in Beijing, China, Atmos. Chem. Phys., 13, 4577-4592, 10.5194/acp-13-4577-2013, 2013.